# END4Rec: Efficient Noise-Decoupling for Multi-Behavior Sequential Recommendation

## ABSTRACT

In recommendation systems, users frequently engage in multiple types of behaviors, such as clicking, adding to cart, and purchasing. Multi-behavior sequential recommendation aims to jointly consider multiple behaviors to improve the target behavior's performance. However, with diversified behavior data, user behavior sequences will become very long in the short term, which brings challenges to the efficiency of the sequence recommendation model. Meanwhile, some behavior data will also bring inevitable noise to the modeling of user interests. To address the aforementioned issues, firstly, we develop the Efficient Behavior Sequence Miner (EBM) that efficiently captures intricate patterns in user behavior while maintaining low time complexity and parameter count. Secondly, we design hard and soft denoising modules for different noise types and fully explore the relationship between behaviors and noise. Finally, we introduce a contrastive loss function along with a guided training strategy to contrast the valid information with the noisy signal in the data, and seamlessly integrate the two denoising processes to achieve a high degree of decoupling of the noisy signal. Sufficient experiments on real-world datasets demonstrate the effectiveness and efficiency of our approach in dealing with multi-behavior sequential recommendation.

## KEYWORDS

Sequential recommendation, Multi-Behavior, Information denoising, Contrastive learning

## 1 INTRODUCTION

With the rapid development of the Internet, recommendation systems have been widely employed on online platforms. Among these, sequential recommendation (SR), predicting the next item for users by regarding historical interactions as temporally-ordered sequences, has attracted various attention from both academia and industry [18, 21, 8, 31, 14, 22, 5].

In reality, users exhibit multiple behaviors when interacting with items, which reflect their multidimensional preferences. For instance, on e-commerce platforms, users can engage with items through various behaviors such as clicking, tagging as favorites, and making purchases. These diverse behaviors represent users' preferences across different dimensions and can serve as auxiliary knowledge to enrich information and enhance the accuracy of recommendation for the target behavior [51, 23, 15, 43, 40, 35].

Recent studies have explored the field of multi-behavior sequential recommendation, such as MMCLR [35], MBSTR [47], S-MBRec [12], and EHCF [2]. Prior research has consistently incorporated multi-behavior information into user representations by directly utilizing transformer or graph neural networks to model different user behaviors respectively. Although the incorporation of multi-behavior information can further effectively explore user

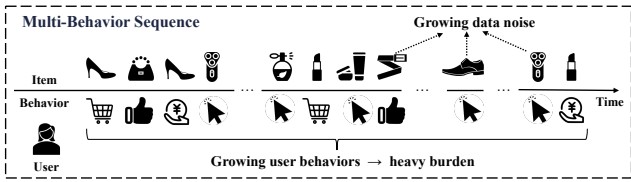

**Figure 1: Illustration of our motivations: increasing user interactions and the resulting large amount of data noise.**

behavior patterns and multidimensional user interest, some behavior data will also bring inevitable noise to the modeling of user interests. For example, as shown in Fig 1, some of the user's clicks may be accidental clicks or unintentional browsing. For example, a woman may have mistakenly clicked on a man's shoe. These noise behaviors bring challenges to the representation of the user's interest. Meanwhile, with diversified behavior data, user behavior sequences will become very long in the short term, which brings challenges to the efficiency of the sequence recommendation model.

In fact, efficiently utilizing user multi-behavior data and adaptively identifying noise data is crucial for achieving more comprehensive representations of user dynamic preferences and generating more accurate sequential recommendation. However, efficient modeling and denoising remain open issues with significant challenges. To begin with, the trade-off between capturing complex patterns in longer multi-behavior sequences and maintaining a manageable model size is a non-trivial problem. Then, noise signals are prevalent and highly coupled with user preference signals in behavior sequences. That brings great challenges to decoupling user preferences from noise due to their close relations and absence of explicit noise annotations. Last but not least, efficiently combining multiple processes such as denoising and representation of user preferences presents a further challenge.

To address these issues, we present a focused study on the efficient noise-decoupling in multi-behavior sequential recommendations. Firstly, we propose an efficient behavior sequence miner (EBM) module. EBM leverages the fast Fourier transform, which has a time complexity of $NlogN$, to replace the complex convolution operation in the time domain. Instead, we use a simpler multiplication operation in the frequency domain, allowing our model to efficiently capture user behavioral patterns with low computational cost. Furthermore, EBM incorporates techniques such as frequency-aware fusion, chunked diagonal mechanism, and compactness regularization to minimize the number of parameters in the model. These methods ensure that while reducing parameter count, our model still maintains its performance levels. Secondly, we divide the user behavior noise into two types: discrete form or token level hard noise, which refers to incorrect clicks made by the user, and continuous form or representation level soft noise, which pertains to outdated user interests. In order to tackle these types of noise effectively, we propose two modules: Hard Noise

Eliminator and Soft Noise Filter. Hard Noise Eliminator considers the correlation between noise and behaviors by considering the preference values of different behaviors during the denoising process, and the Soft Noise Filter places different behaviors in separate channels for denoising. To fully separate user interest from noise in our data, we introduce a technique called Noise-Decoupling Contrastive Learning. This approach aims at removing noise effectively while preserving important user interests. Finally, to effectively combine the denoising processes, we propose a guided training strategy consisting of four steps: pre-training, hard noise comparison, soft noise comparison, and final convergence. This strategy seamlessly integrates both denoising processes to enhance the decoupling of noise signals and improve the overall denoising effect. By gradually improving the model's ability to handle noise signals, this strategy also enhances the robustness of the training process.

To summarize, the contributions of this article are as follows:

- We present a focused study from a novel noise-decoupling perspective in sequential recommendation.
- We propose Efficient Behavior Sequence Miner (EBM), which can adequately capture complex user patterns while maintaining low model complexity and parameter counts by exploiting frequency-aware fusion, chunking diagonal mechanism, and compactness regularization.
- We propose a Behavior-Aware Denoising module including Hard Noise Eliminator at the discrete token level, Soft Noise Filter in the continuous representation space, Noise-Decoupling Contrastive Learning, and a guided training process to achieve effective noise removal.
- We conducted comprehensive experiments on three real-world datasets. The experimental results demonstrate the effectiveness of END4Rec, and complexity analysis proves its efficiency.

## 2 RELATED WORK

### 2.1 Multi-Behavior Sequential Recommendation

Recommendation system recommends personalized content based on individual preferences, sequential recommendation predicts a user's next target item based on their historical behavior, playing a crucial role in enhancing the user experience on online platforms. With the emergence of deep learning, sequential recommendation models such as BERT4Rec [26], DIN [52], SASRec [19], and FEARec [9] were introduced for recommendation tasks. However, they failed to consider the diversity of user interactions in real-world scenarios, such as clicking, liking, and purchasing in e-commerce, which provided valuable insights into user intent. To overcome this limitation, researchers have proposed various methods for handling multi-behavior data.

Previous research has explored the use of multi-task frameworks to optimize recommendation systems. One approach is to model the cascade relationship among different user behaviors, as done in NMTR [10]. Another approach is to assign user behaviors to distinct tasks and employ hierarchical attention mechanisms to improve recommendation efficiency, as in DIPN [13]. Other studies have focused on enhancing recommendation by fusing multi-behavior data and using other behaviors as auxiliary signals. This has been

achieved through attention mechanisms [34, 44], graph neural networks [3, 37], or other related approaches. For example, MATN [41] used a transformer and gated network to capture behavior relationships, while CML [32] introduced a multi-behavior contrastive learning framework to enhance behavior representations. KMCLR [43] utilized comparative learning tasks and functional modules to improve recommendation performance through the integration of multiple user behavior signals.

Although the fusion of multi-behavior information can further effectively explore user behavior patterns and multi-dimensional user interests, some behavior data will inevitably introduce noise to the modeling of user interests. This noise poses challenges to the judgment of user sequence interests. Moreover, with the diversification of behavior data, user behavior sequences become increasingly lengthy in a short period, which presents challenges to the efficiency of the sequence recommendation model.

### 2.2 Denoising in Recommendation

Earlier studies have employed user explicit feedback to reduce the gap between implicit feedback and user preference [4, 20, 50], but in real-world scenarios, acquiring user feedback has become increasingly challenging, with usually very few users willing to spend time providing evaluations for products. As a result, the denoising problem has gradually evolved into an unsupervised challenge, prompting numerous studies to approach noise determination from various perspectives.

One line is to remove the discrete noise by removing the item from the sequence, which is also known as hard denoising. For instance, CLEA [25] determines the noisy items based on the target items and divides the items in the basket into positive and negative sub-baskets. In contrast, RAP [28] formulates denoising as a Markov Decision Process (MDP) and learns a policy network to guide an agent in deciding whether to remove items, thereby explicitly eliminating irrelevant elements within the sequence. ADT [30] introduces an adaptive denoising training strategy to reduce noise, because noisy feedbacks typically have high loss in the early stages of training. In addition, BERD [27] conducted an integration of these high-loss instances with uncertainty measurements to distinguish unreliable instances. The other line is to remove continuous noise by removing it from the representation level or feature level, which is also known as soft denoising. For example, DSAN [48] introduces the use of the max function to automatically eliminate attention weights for irrelevant items by considering a virtual target item. FMLP-Rec [53] treats sequence representations as signals and further incorporates Fast Fourier Transform (FFT) and learnable filters to learn better sequence representations. Furthermore, DPT [49] introduces a three-stage paradigm involving de-noising and prompt fine-tuning, progressively mitigating the impact of noise through data-driven processes. However, previous denoising methods did not perform noise determination from the perspective of overall sequence perception, as well as did not fully consider the diversity of noise types and their relationship with user behavior, resulting in unsatisfactory results.

## 3 PROBLEM DEFINITION

In recent years, the problem of sequential recommendation has gained significant attention in the field of recommendation systems.

However, most existing works have focused on general sequential recommendation, which predicts the next item based on single-type interaction sequences, ignoring the multi-type user behaviors. In real-world scenarios, such as e-commerce platforms, users often interact with items through different behaviors like *clicking*, *adding to favorites*, *adding to cart*, and *purchasing*, reflecting diverse preferences. To address this limitation, we study the problem of multi-behavior sequential recommendation, which aims to model the complex relationships between different user behaviors and transfer general preferences to the targeted behavior for the recommender's decision. We define the problem as follows:

DEFINITION. **(Multi-Behavior Sequential Recommendation)** *Given the sets of users $U$, items $V$, and types of behavior $B$, for a user $u$ ($u \in U$), his/her behavior-aware interaction sequence $S_u$ consists of individual triples $(v, b, p)$ which are ordered by time. Each triple represents the interacted item $v$ under the behavior type $b$ at position $p$ in the sequence. Thus, the input of the problem is the behavior-aware interaction sequence $S_u = [(v_1, b_1, p_1), (v_2, b_2, p_2), ..., (v_L, b_L, p_L)]$ of the user $u$ and the output $(v_{L+1}, b_t, p_{L+1})$ is the predicted next item $v_{L+1}$ of the targeted behavior $b_t$ at next position $L + 1$.*

## 4 METHODOLOGY

### 4.1 Overview

In this section, we introduce our proposed END4Rec model, which is able to efficiently mine user multi-behavior sequences with $O(N \log N)$ complexity and fully decouple the noise in the sequences. Specifically, we first introduce Behavior-Aware Sequence Embedding (4.2), which fuses item information, behavior information, and location information to help the model understand user behavior sequences more comprehensively. Then, to improve the efficiency of long behavioral sequence model mining, we design an efficient base module called Efficient Behavior Sequence Miner (EBM) (4.3), which can efficiently mine user behavior patterns with low model complexity by exploiting frequency-aware fusion, chunking diagonal mechanism, and compactness regularization. Based on the high efficiency of EBM, we are able to realize the overall perceptual denoising of sequences. Further, considering different types of noise signals and user behaviors, we propose a Behavior-Aware Denoising module including Hard Noise Eliminator at the discrete token level (Here token is each triple $(v, b, p)$ in the input sequence.) and soft Noise Filter in the continuous representation space (4.4). Finally, in order to better realize noise decoupling, we introduce Noise-Decoupling Contrastive Learning and a guided training process to achieve effective noise removal (4.5). The overall structure and flow of our model are visually represented in Figure 2.

### 4.2 Behavior-Aware Sequence Embedding

The embedding layer of END4Rec integrates item information ($v$), behavior information ($b$), and position information ($p$). For a triad $(v, b, p)$ within a user behavior sequence ($S$), the embedding is denoted as follows:

$$e = e_v + e_b + e_p, \quad S = [e_1, e_2, ..., e_L] \in \mathbb{R}^{L \times d}, \quad (1)$$

where $L$ represents the sequence length, and $d$ denotes the embedding size. This embedding combines item information, behavior information, and position information to reflect the user's behavioral sequences more comprehensively, which helps to improve the model's understanding of the user's interests and behaviors, and thus improves the accuracy of personalized recommendation.

### 4.3 Efficient Behavior Sequence Miner

In order to fully exploit the temporal characteristics of multiple user behaviors, we often splice various user behaviors into a sequence according to timestamps. As the number of user interactions increases, the length of the sequence grows, which challenges the model's efficiency.

To address the aforementioned challenge, we draw on the convolution theorem [6], which shows that the product operator in the frequency domain is equivalent to the convolution operator in the time domain. This means that we can realize complex convolution operations by fast Fourier transforms with $N logN$ time complexity as well as multiplication operations. Specifically, we transform the user behavior sequence $S$ into the frequency domain ($X$) with the help of Fourier Transform [53] and then realize the convolutional fusion of the complex user behavior tokens by the dot product operation. Finally, the fully convolved sequence representation $\widetilde{S}$ is obtained by inverse transformation. Due to the $O(N \log N)$ computational complexity of the Fast Fourier Transform algorithm [24], this process is able to improve efficiency while fully exploiting the user's behavioral patterns. The specific formula is as follows:

$$X = \mathcal{F}(S) \in \mathbb{C}^{L \times d}, \quad \widetilde{X} = W \odot X, \quad \widetilde{S} \leftarrow \mathcal{F}^{-1}\left(\widetilde{X}\right) \in \mathbb{R}^{L \times d}, \quad (2)$$

where $S$ represents the user behavior sequence, $X$ represents the frequency domain representation of $S$, $\widetilde{S}$ represents the convolved sequence features, $W$ denotes the dot product matrix and $\mathbb{C}$ denotes the complex space. However, this approach encounters three challenges. First, the dot product operation cannot fully integrate the information of various frequency bands of the model (manifested in the time domain as the user's interest information in multiple time scales). Second, the number of parameters of $W$ grows with the increase of the length of the input sequences, and it is difficult for the model to flexibly adapt to the change of the input length. Third, in our behavior-modeling situation, user behavior is easily concentrated in a certain frequency band [53], and the model can easily overfit local information.

To this end, we propose the Efficient Behavior Sequence Miner (EBM) and enhance three key aspects of the dot-product operation involving the $W$-matrix within it. First, we utilize matrix multiplication instead of dot product to achieve better frequency domain fusion however, this will further increase the number of model parameters. So we further propose Chunked Diagonal Mechanism, which enables model parameters to be shared among different tokens so that the model can adaptively handle sequences of different lengths. Finally, considering that user behavior information may cluster in certain frequency bands, we design a Compactness Regularization method for sparsifying the tokens in the frequency domain. The specific process is as follows:

***Frequency-Aware Fusion.*** In EBM, the first improvement involves utilizing matrix multiplication instead of the traditional dot product operation, i.e., $W \in \mathbb{C}^{L \times d \times d}$ instead of $W \in \mathbb{C}^{L \times d}$. This approach

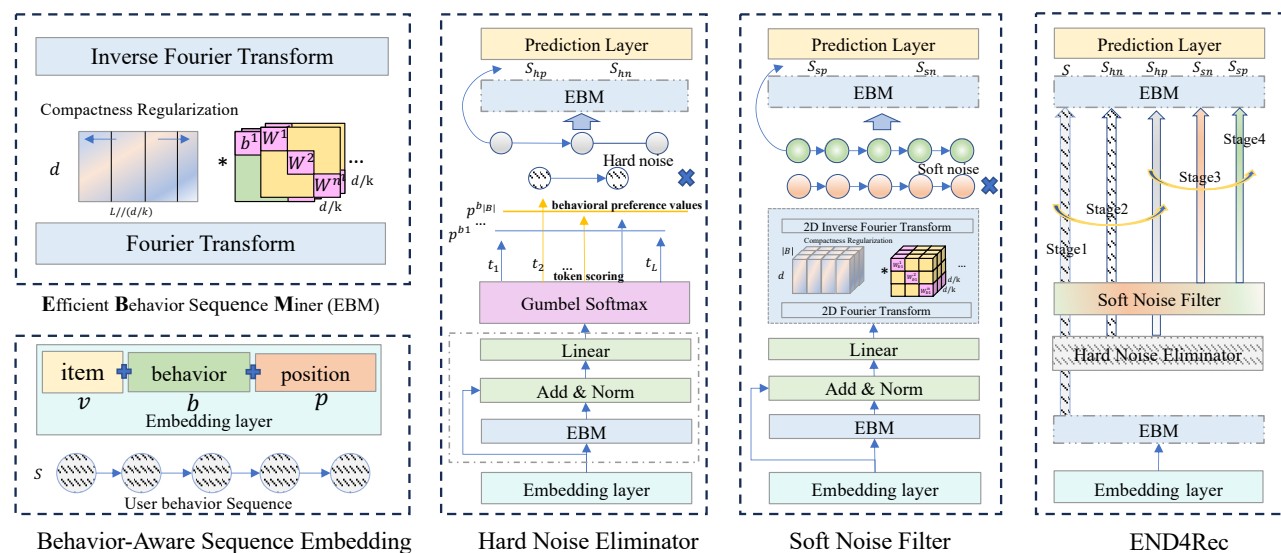

**Figure 2: Our END4Rec model, featuring an efficient user multi-behavior sequence mining process with $O(NlogN)$ complexity and adaptive input length. The framework consists of several key components: Efficient Behavior Sequence Miner (EBM) (4.3) for mining behavior patterns efficiently, Behavior-Aware Denoising (4.4) including Hard Noise Eliminator and Soft Noise Filter for noise detection, and Noise-Decoupling Contrastive Learning (4.5) for sufficient noise decoupling.**

allows for the effective fusion of user behavioral information across different frequency bands. By considering long and short-term user interests in multiple time scales, EBM allows for more efficient mining of user behavior patterns.

***Chunked Diagonal Mechanism.*** To address the issue of increasing matrix parameters as user sequences grow, EBM introduces a chunked diagonal mechanism for complex weight matrices. The weight matrix $W \in \mathbb{C}^{L \times d \times d}$ is decomposed into $k$ shared weight matrices $W^n \in \mathbb{C}^{d/k \times d/k}$ ($n = 1, \ldots, k$), each with reduced dimensions. This decomposition into $k$ smaller diagonal weight matrices, denoted as $W^n$, is somewhat interpretable, similar to $k$-head attention, while enabling computational parallelization. So we get $\tilde{x}_i^n = W^n x_i^n$, where $x_i^n$ represents the $n$-th block of the $i$-th frequency token ($i \in [1, L//(d/k)]$). Specifically, we employ a double-layer MLP structure as $W^n$. The formula is as follows:

$$\tilde{x}_i^n = \text{MLP}\left(x_i^n\right) = W_2^n \sigma \left(W_1^n x_i^n + b_1^n\right) + b_2^n, \quad (3)$$

where the weights $W^n$ and $b^n$ are shared are all tokens, and thus the parameter count can be significantly reduced. ***Compactness Regularization for Token Sparsity***. User behavior is easily concentrated in a certain frequency band [49], resulting in behavioral features not being adequately fused in the frequency domain, and the model can easily overfit local information, so EBM incorporates a regularization loss that promotes the sparsity of tokens in the frequency domain. Compactness is a desired trait of intra-factor representations and its opposite is what we expect for inter-factor representations. ReduNet [1] proposed to measure compactness of representation with rate-distortion $R(z, \epsilon)$, which determines the minimal number of bits to encode a random variable $z$ subject to a decoding error upper bounded by $\epsilon$. Inspired by this, we design a compactness regularization loss function to control the diversity of

the token space. The specific formula is as follows:

$$R(\mathcal{X}, \epsilon) = \frac{1}{2} \log \det \left(I + \frac{d}{L\epsilon^2} \mathcal{X} \mathcal{X}^T\right), \quad (4)$$

where $\mathcal{X} \in \mathbb{C}^{L//(d/k) \times d}$ is the token matrix and the rest are hyperparameters. log det means the logarithm of the determinant of a matrix and $I$ is the identity matrix. By introducing a representation compactness metric and a corresponding regularized loss function Eq. (10), tokens are sparsified during the training process to control their spatial diversity for better fusion of behavioral features.

In conclusion, our proposed Efficient Behavior Sequence Miner (EBM) can both capture complex user patterns adequately and maintain model simplicity through the utilization of frequency-aware fusion, the chunking diagonal mechanism, and compactness regularization. What's more, the model efficiency analysis will be given in the experiments.

## 4.4 Behavior-Aware Denoising Module

The growth of user behavior sequences often introduces significant data noise, which can be categorized into discrete forms (like incorrect clicks or implicit negative reviews) and continuous forms (reflecting outdated user interests). Prior denoising methods [53, 28] fail to take into account the different types of noise signals, as well as ignore the differences between different noise signals and their relationships with the types of user behaviors, thus leading to sub-optimal results of their algorithms. Additionally, the noise present in the behavior sequence typically derives from the overall behavior of the user. However, previous methods [25, 30] face limitations in handling this complex situation due to computational complexity. Furthermore, these methods often rely on certain assumptions when determining the noise, which may not be valid in

many cases. As a result, achieving a satisfactory decoupling of the noise signal becomes a challenging task.

We propose EBM to better solve the efficiency problem and provide a basis for realizing the overall perceptual denoising. We design a Hard Noise Eliminator for discrete noise and a Soft Noise Filter for continuous noise based on the EBM module by further considering the noise types and behavioral properties. At the same time, we also design two contrast loss functions to realize the complete decoupling of the noise signal from the user preference.

*4.4.1* **Hard Noise Eliminator**. Given a sequence of user behaviors, we further design a Hard Noise Eliminator (HNE) on top of EBM's efficient mining of user behavioral relationships. HNE discriminates noise tokens by overall perceived token scoring $t_i$ and behavioral preference values $p_b$.

To mine the association between hard noise and different behaviors, we consider assigning different preference values $p_b$ for different behavior types $b$. Specifically, since different behaviors occur at varying frequencies and are influenced by different user interests, we model different behavioral preference values with Poisson Distribution [7]. Poisson Distribution can represent how often a user performs different behaviors over a period of time, thus reflecting the degree of user interest in various behaviors [16, 11]. We simply approximate the preference values of different behaviors with the peaks of the Poisson Distributions of different $\lambda_b$, which can be searched as a hyperparameter, and the purpose of adding 1 is to make it easier to set the threshold later.

$$p_b = P\{X = \lambda_b\} + 1 = \frac{e^{-\lambda_b} \cdot \lambda_b^{\lambda_b}}{\lambda_b!} + 1. \tag{5}$$

In order to obtain the overall perceptual token score $t_i$, we add a residual layer to the EBM framework, an addition that enhances the training process of the model, making it more manageable and stable, and a linear layer as well as sigmoid activation.

Finally, for $v_i$ in sequence, we split the original sequence into two mutually exclusive sequences by treating the token $p_{b_i} - t_i < 0.5$ as a noise signal. However, this hard coding is not differentiable and prevents the model from being trained well via back-propagation. To address this issue, inspired by [29, 46], we integrate Gumbel Softmax into our denoising generator as a differentiable surrogate to support model learning over the discrete output. Specifically, the new denoising generator is rewritten as follows:

$$\mathcal{J}(v_i) = \frac{\exp\left(\left(\log(p_{b_i} - t_i) + g_1\right)/\tau\right)}{\sum_{y=0}^{1} \exp\left(\log\left((p_{b_i} - t_i)^y(1 - (p_{b_i} - t_i))^{1-y}\right) + g_y\right)/\tau\right)}, \tag{6}$$

where $g_y$ is i.i.d sampled from a Gumbel Distribution, serving as a noisy disturber: $g = -\log(-\log(x))$ and $x \sim Uniform(0, 1)$. $\tau$ is the temperature parameter to smooth the discrete distribution. We classify them into filtered and noisy sequences according to $\mathcal{J}(v_i)$ and obtain the hard denoised sequence representation $S_{hp}$ and the hard noise sequence representation $S_{hn}$ through the same EBM.

*4.4.2* **Soft Noise Filter**. After removing the discrete token-level noise from the user's behavioral sequences by Hard Noise Eliminator, below we consider the filtering of soft noise at the representation or feature level. Some related studies have shown that the

noise information in the sequence of user behaviors can be eliminated through a learnable filter kernel in the frequency domain [53]. Here, we utilize the $W^n$ matrices in the EBM as the learnable filtering kernel. Further, in order to enhance the model's ability to distinguish between different behaviors, we map different types of behaviors to different channels to obtain the EBM+ module and extract the difference to obtain the soft noise signal and the filtered signal. The formula is shown below:

$$\tilde{x}_{i,b}^n = W_b^n x_{i,b}^n, \quad n = 1, \ldots, k, \tag{7}$$

where $W_b^n$ represents the $n$th block of complex matrices for behavior $b$ channel. Finally we get the filtered information $\widetilde{X}$ and and the noise $X - \widetilde{X}$. Thus, similarly, we can obtain the soft-filtered sequence representation $S_{sp}$ and the soft-noise sequence representation $S_{sn}$ by the same EBM.

## 4.5 Noise-Decoupling Contrastive Learning

We obtain a set of denoised and noisy sequences by Hard Noise Eliminator and Soft Noise Filter respectively. In order to make up for the insufficiency of the supervised signals, and at the same time to realize better noise decoupling, we make the following reasonable assumptions: (1) the effect of denoised sequences is better than that of the original sequences, and (2) the effect of the original sequences is better than that of the noise sequences. Specifically, the following contrastive loss is designed for noise decoupling contrastive learning, which is calculated as follows:

$$Q(S) = \frac{\exp\left(S \cdot e_{v_t}\right)}{\sum_{v \in \mathcal{V}} \exp\left(S \cdot e_v\right)}, \tag{8}$$

$$\mathcal{L}_{CL}(S_p, S, S_n) = -\sum_{\{v_t\}} \left[\log \sigma\left(Q\left(S_p\right) - Q\left(S\right)\right) + \log \sigma\left(Q\left(S\right) - Q\left(S_n\right)\right)\right], \tag{9}$$

where $e_v$ is the item embedding of $v$, the $v_t \in \{v_{L+1}\}$ is the target item to be predicted, and $S_p, S, S_n$ are the representation of the denoised sequence original sequence and the noise sequence, respectively. Maximizing $Q(S)$ is the goal of our optimization. In addition, due to the sparse supervised signals, in order to better guide the model to learn noise decoupling, we design the following guided training process to facilitate the optimization of END4Rec. To illustrate the process, the following loss function is first defined, where $\mathcal{L}_{compactness}$ is the regular loss of the EBM.

$$\mathcal{L}_{compactness} = \sum_{u_i \in U} R(X_i, \epsilon), \quad \mathcal{L}_{pred} = -\sum_{u_i \in U} Q(S_{u_i}). \tag{10}$$

For the sake of simplicity in illustration, we omit the regular loss of the model parameters and the hyperparameters in front of each loss function. The specific training process is presented below.

As shown in Algorithm 1, we randomly initialize all parameters. Then, we use the method of freezing parameters to gradually train the EBM layer, hard denoising layer, and soft denoising layer of the model, and make the noise signal continuously decoupled by continuously adding contrastive loss. We will show that better results can be achieved than end-to-end training in ablation experiments.

---

**Algorithm 1** Noise-Decoupling Contrastive Learning

---
**Input**: Data and Hyperparameters
**Output**: Model parameters $\Phi(\Phi_{EBM}, \Phi_{hard}, \Phi_{soft})$
Random initialization $\Phi$
**Stage 1**: Training the embedding layer and EBM layer
    Calculate loss: $\mathcal{L}_1 = \mathcal{L}_{pred}(S) + \mathcal{L}_{compactness}$
    Update parameters $\Phi_{EBM}$ to minimize $L_1$
**Stage 2**: Training the Hard Noise Eliminator
    Split $S$ into $S_{hp}$ and $S_{hn}$ according to Eq. (6)
    Calculate loss:
        $\mathcal{L}_2 = \mathcal{L}_{pred}(S_{hp}) + \mathcal{L}_{CL}(S_{hp}, S, S_{hn}) + \mathcal{L}_{compactness}$
    Update parameters $\Phi_{EBM}, \Phi_{hard}$ to minimize $\mathcal{L}_2$
**Stage 3**: Training the Soft Noise Filter
    Split $S_{hp}$ into $S_{sp}$ and $S_{sn}$ according to Eq. (7)
    Calculate loss:
        $\mathcal{L}_3 = \mathcal{L}_{pred}(S_{sp}) + \mathcal{L}_{CL}(S_{hp}, S, S_{hn}) + \mathcal{L}_{CL}(S_{sp}, S_{hp}, S_{sn})$
        $+\mathcal{L}_{compactness}$
    Update parameters $\Phi_{EBM}, \Phi_{hard}, \Phi_{soft}$ to minimize $\mathcal{L}_3$
**Stage 4**: Train the final model to converge
    Fix $\Phi_{hard}, \Phi_{soft}$ and get $S_{sp}$ through the denoising module.
    Calculate loss: $\mathcal{L}_4 = \mathcal{L}_{pred}(S_{sp}) + \mathcal{L}_{compactness}$
    Update model parameters $\Phi_{EBM}$ to minimize $\mathcal{L}_4$

---

## 5 EXPERIMENT

### 5.1 Experimental Setting

*5.1.1 Datasets.* In order to evaluate the performance of END4Rec, we conduct experiments on three large-scale real-world recommendation datasets that are widely used in multi-behavioral sequential recommendation research and are considered standard benchmarks[45, 38]. These datasets contain various user interaction behaviors, including *clicking*, *adding to favorites*, *adding to cart*, and *purchasing*. Specifically, the IJCAI dataset was released by the IJCAI Contest 2015 for the task of predicting repeat buyers. The CIKM and Taobao datasets were released by E-commerce companies Alibaba and Taobao, which are the largest online communication platforms in China.

For the data processing, we follow a similar approach to previous studies (e.g., [19, 26]) by removing users and items with fewer than 20 interaction records, which ensures that the users and items in the dataset had sufficient interaction data to accurately capture their preferences and behaviors. Next, we focus on the *purchase* behavior as it directly benefits the platforms financially. To ensure sufficient representation of this behavior in each user sequence, we require it to appear at least 5 times [49]. In addition to the above steps, we also perform general data deduplication and cleaning to enhance the stability and reproducibility of the experimental results. The detailed statistical information of processed datasets is summarized in Table 1.

*5.1.2 Comparison Baselines.* To evaluate the effectiveness of the proposed method END4Rec, we conduct a comprehensive comparison with several state-of-the-art baselines as follows:

**General Sequential Recommendation Methods. SASRec** [19] utilizes a transformer-based encoder to learn sequence and item representations. **BERT4Rec** [26] employs a bidirectional encoder

**Table 1: Statistical information of experimental datasets.**

| Dataset | CIKM | IJCAI | Taobao |
|---|---|---|---|
| #users | 254,356 | 324,859 | 279,052 |
| #items | 521,900 | 331,064 | 731,517 |
| #interactions | 18,824,670 | 46,694,666 | 32,758,555 |

with transformers to model sequential information and is optimized using the Cloze objective. **CL4SRec** [42] combines contrastive SSL with a Transformer-based SR model, incorporating crop, mask, and reorder augmentation operators. **FEARec** [9] improves the original time domain self-attention in the frequency domain with a ramp structure, allowing for the explicit learning of both low-frequency and high-frequency information.

**Multi-Behavior Recommendation Methods. MB-GCN** [17] is a graph-based model designed to address the issue of data sparsity in multi-typed user behavior data modeling. This approach employs graph convolution operations for message passing. **KHGT** [39] introduces a transformer-based approach for multi-behavior modeling, with a focus on temporal information and auxiliary knowledge incorporation. The model utilizes graph attention networks to capture behavior embeddings. **CML** [32] integrates contrastive learning into multi-behavior recommendation by proposing meta-contrastive coding. This enables the model to learn personalized behavioral features. **KMCLR** [43] proposes a framework that enhances recommender systems through two comparative learning tasks and three functional modules: multiple user behavior learning, knowledge graph enhancement, and coarse- and fine-grained modeling of user behaviors to improve performance.

**Denoising Methods for Recommendation. CLEA** [25] utilizes a discriminator to divide a basket into positive and negative sub-baskets based on noise detection. **ADT** [30] adaptively prunes noisy interactions to achieve implicit feedback denoising. **FMLP-Rec** [53] incorporates Fast Fourier Transform (FFT) and inverse FFT procedures to minimize the influence of noise and improve sequence representations. **DPT** [49] introduces a three-stage paradigm that involves de-noising and prompt fine-tuning, progressively mitigating the impact of noise through data-driven processes. In accordance with previous studies [45, 36], we employ these denoising baselines on the multi-behavior problem by incorporating behavioral types into the input embedding to ensure a fair comparison.

*5.1.3 Evaluation Metrics.* In this study, we assess the performance of comparison methods for the top-K recommendation using two evaluation metrics: Hit Ratio (HR@K) and Normalized Discounted Cumulative Gain (NDCG@K). HR@K measures the average proportion of relevant items in the top-K recommended lists, while NDCG@K evaluates the ranking quality of the top-K lists in a position-wise manner. To ensure fair and efficient evaluation, each positive instance is paired with 99 randomly selected non-interactive items identical to recent state-of-the-art works [39, 45, 53, 38]. We employ the leave-one-out strategy for performance evaluation [33], where each user's temporally ordered last purchase serves as the test sample and the previous one as the validation sample.

*5.1.4 Implementation Details.* To ensure a fair comparison between different models, we conduct consistent settings across all methods. Specifically, we set the embedding size to 128 and the batch size to

**Table 2: Overall performance comparison of all methods in terms of HR@K and NDCG@K (K=10, 20). (p-value < 0.05)**

| Datasets | CIKM | | | | Taobao | | | | IJCAI | | | |
|---|---|---|---|---|---|---|---|---|---|---|---|---|
| Metric | H@10 | N@10 | H@20 | N@20 | H@10 | N@10 | H@20 | N@20 | H@10 | N@10 | H@20 | N@20 |
| SASRec | 0.3055 | 0.1861 | 0.3981 | 0.2285 | 0.2995 | 0.1766 | 0.3834 | 0.2129 | 0.3505 | 0.1878 | 0.4553 | 0.2346 |
| Bert4Rec | 0.3350 | 0.2147 | 0.4235 | 0.2337 | 0.2856 | 0.1670 | 0.3679 | 0.1991 | 0.3748 | 0.2060 | 0.4890 | 0.2658 |
| CL4Rec | 0.3422 | 0.2140 | 0.4392 | 0.2472 | 0.3132 | 0.1840 | 0.4022 | 0.2206 | 0.3877 | 0.2104 | 0.5047 | 0.2672 |
| FEARec | 0.3392 | 0.2093 | 0.4386 | 0.2493 | 0.3213 | 0.1891 | 0.4120 | 0.2274 | 0.3866 | 0.2085 | 0.5027 | 0.2626 |
| MBGCN | 0.3889 | 0.2136 | 0.4951 | 0.2894 | 0.3628 | 0.2016 | 0.4510 | 0.2498 | 0.3627 | 0.1969 | 0.4721 | 0.2502 |
| KHGT | 0.4014 | 0.2305 | 0.5160 | 0.2993 | 0.3824 | 0.2181 | 0.4815 | 0.2670 | 0.4104 | 0.2214 | 0.5337 | 0.2791 |
| CML | 0.4234 | 0.2466 | 0.5400 | 0.3060 | 0.4212 | 0.2420 | 0.5311 | 0.2942 | 0.4344 | 0.2366 | 0.5658 | 0.3018 |
| KMCLR | 0.4344 | 0.2494 | 0.5584 | 0.3239 | 0.3587 | 0.2070 | 0.4563 | 0.2503 | 0.4441 | 0.2397 | 0.5769 | 0.3051 |
| CLEA | 0.4278 | 0.2349 | 0.5446 | 0.3184 | 0.3990 | 0.2218 | 0.4961 | 0.2747 | 0.3989 | 0.2166 | 0.5194 | 0.2752 |
| ADT | 0.2536 | 0.1463 | 0.2926 | 0.1688 | 0.2302 | 0.1307 | 0.2519 | 0.1463 | 0.2787 | 0.1541 | 0.3436 | 0.1956 |
| FMLP | 0.3764 | 0.2354 | 0.4831 | 0.2719 | 0.3446 | 0.2024 | 0.4424 | 0.2427 | 0.4265 | 0.2315 | 0.5552 | 0.2940 |
| DPT | 0.4431 | 0.2545 | 0.5697 | 0.3304 | 0.4221 | 0.2408 | 0.5315 | 0.2947 | 0.4531 | 0.2445 | 0.5709 | 0.3082 |
| **END4Rec** | **0.4787** | **0.2754** | **0.6120** | **0.3536** | **0.4464** | **0.2533** | **0.5614** | **0.3102** | **0.4821** | **0.2613** | **0.6166** | **0.3314** |
| Improvement | 8.02% | 8.23% | 7.44% | 7.00% | 5.75% | 4.71% | 5.63% | 5.27% | 6.38% | 6.88% | 6.88% | 7.53% |

512. During the experimentation process, we employ a grid search approach to identify the optimal hyperparameters for each model. The maximum number of epochs is set to 1000, and the training process is halted if the NDCG@K summation on the validation dataset has not improved for 20 consecutive epochs.

## 5.2 Overall Performances

**Effectiveness Analysis**. In Table 2, we present a comprehensive performance comparison across different datasets and summarize the following observations: (1). We observe that multi-behavioral approaches typically outperform general sequential recommendation approaches, which underscores the significance of our underlying research problem. By modeling the complex relationships between different user behaviors with the aid of multiple behavioral data as auxiliary information, we effectively translate general preferences to targeted behaviors to enhance performance. (2). Partial denoising methods outperform multi-behavior sequential recommendation, confirming our research motivation that multi-behavioral sequences contain significant amounts of data noise that can be effectively reduced using denoising methods. (3). The performance of different types of denoising methods or even the same denoising algorithm on different datasets varies greatly, suggesting that many methods have limitations based on specifically targeted items or loss functions. These limitations ignore the diverse types of noises hidden in multiple user behaviors, making it difficult to obtain consistent results compared to END4Rec, which has more general assumptions. (4). END4Rec consistently outperforms other baselines on all datasets, demonstrating its effectiveness and generalizability. This is due to the fact that the discrimination of noise depends not only on overall user behavior preferences but also considers the type of noise and its connection with different behaviors. Moreover, the proposed contrastive learning and gradual training strategy further aid in efficiently decoupling noise from complex multi-behavior data.

**Efficiency Analysis**. To further illustrate the core efficient module EBM in END4Rec, we conduct a complexity comparison analysis with some representative methods, such as Self-Attention and FMLP [53], as shown in Table 3. From the table, we can conclude that both END4Rec and FMLP have lower complexity compared to traditional self-attention methods. Although FMLP is based on

**Table 3: Complexity, parameter count, and degree of feature fusion for SA, FMLP, and EBM. $L$, $d$, and $k$ refer to the sequence length, hidden size, and block count, respectively.**

| Model | Complexity | Parameter Count | Feature fusion |
|---|---|---|---|
| Self-Attention | $L^2d + 3Ld^2$ | $3d^2$ | Adequate |
| FMLP | $Ld + LdlogL$ | $Ld$ | Inadequate |
| EBM | $Ld^2/k + LdlogL$ | $(1 + 4/k)d^2 + 4d$ | Adequate |

Fourier Transform operation to accelerate efficiency, it only utilizes the dot-product operator and fails to fuse frequency domain information, which cannot model more complex patterns in behavior sequences. In contrast, END4Rec with the aid of EBM can achieve full fusion (The ability to fully intersect global frequency domain features.) while requiring relatively small parameters, independent of behavior sequence length. This ensures the efficiency of END4Rec in terms of both running time and parameter spaces.

## 5.3 Ablation Study

To investigate the effectiveness of each component, we introduce the following variants of END4Rec: ***Variant without Hard Noise Eliminator (w/o hard)*** and ***Variant without Soft Noise Filter (w/o soft)*** represent that we discard the hard or soft denoising module and the corresponding contrastive loss and training process in END4Rec, respectively. ***Variant without Noise-Decoupling Contrastive Learning (w/o cl)*** indicates that we do not use two contrastive loss functions during the training process. ***Variant without Compactness Regularization (w/o camp)*** indicates that the compactness regularization loss is not used in all EBM modules. ***Variant without Guide Optimization (w/o opti)*** uses end-to-end training methods instead of guided training methods.

The comparison results are presented in Table 4: First, the (***w/o hard***) and (***w/o soft***) variants demonstrate the effectiveness of our proposed combination of soft and hard denoising methods, where different datasets are affected by the two types of noise to varying degrees, and the combination of the two types of methods achieves a better result. Second, the (***w/o cl***) variant demonstrates the learning effectiveness of our proposed noise decoupling comparison. It mitigates the inadequacy of supervised signals and decouples noise from the data. Third, (***w/o camp***) variant demonstrates the superiority of compactness regularization, which can control the sparsity of

**Table 4: Ablation study with key modules in END4Rec.**

| Datasets | CIKM | | Taobao | | IJCAI | |
|---|---|---|---|---|---|---|
| Metric | H@10 | N@10 | H@10 | N@10 | H@10 | N@10 |
| *w/o hard* | 0.3614 | 0.2260 | 0.3308 | 0.1943 | 0.4094 | 0.2222 |
| *w/o soft* | 0.4107 | 0.2255 | 0.3831 | 0.2129 | 0.3830 | 0.2079 |
| *w/o cl* | 0.4254 | 0.2443 | 0.4052 | 0.2311 | 0.4350 | 0.2347 |
| *w/o comp* | 0.4387 | 0.2519 | 0.3623 | 0.2090 | 0.4486 | 0.2421 |
| *w/o opti* | 0.4576 | 0.2590 | 0.4263 | 0.2432 | 0.4577 | 0.2470 |
| **END4Rec** | **0.4787** | **0.2754** | **0.4464** | **0.2533** | **0.4821** | **0.2613** |

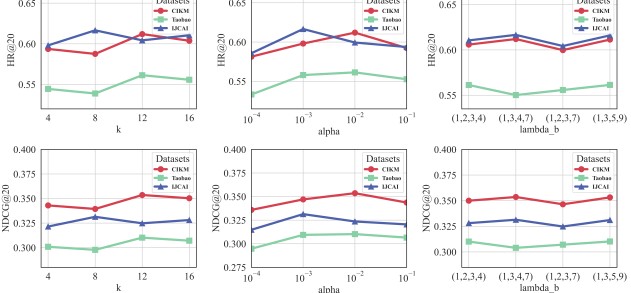

**Figure 3: Hyper-parameter study of the END4Rec.**

the frequency domain token and prevent the model from overfitting to local information. Finally, (*w/o opti*) variant demonstrates the effectiveness of our guided training process, which, through four steps, is able to guide the model step-by-step to decouple the noise from the data and achieve the optimal result.

### 5.4 Hyper-parameter Analysis

To investigate the effects of hyper-parameters in END4Rec, we perform experiments with different hyper-parameter configurations and present results in Fig 3, we can conclude as follows: *(1) Number of matrix blocks of Chunked Diagonal Mechanism $k$.* we find that $k$ values ranging from 8 to 12 yield optimal results. However, there is a noticeable decrease in performance as the value of k increases beyond this range, which may be attributed to the limited feature fusion capabilities for each matrix block resulting in smaller size blocks. *(2) Compactness regularization parameter $\alpha$.* We observe that an appropriate range of values for $\alpha$, between 0.01 and 0.001, achieves better performance. It suggests that excessive regularization loss can negatively impact the normal representation of vectors, while insufficient regularization may not adequately constrain the sparse representation of frequency domain tokens. *(3) Hyper-parameters for each behavioral preference value $\lambda_b$.* We sort the behaviors in the dataset according to the frequency order of the behaviors in the dataset, which are *adding to favorites*, *purchasing*, *adding to cart*, and *clicking*, and take four sets of hyperparameters [(1, 2, 3, 4), (1, 3, 4, 7), (1, 2, 3, 7), (1, 3, 5, 9)], where each tuple within the parentheses means the $\lambda_b$ values for four distinct behaviors, and the experimental results find that the parameter's sensitivity is not high, which indicates that the model is able to converge to different ranges according to different thresholds.

### 5.5 Visualization Analysis

To better illustrate the intricate connection between noises and behaviors, we conduct visual experiments depicted in Fig 4 and observe the following points: (1). For the hard denoising process,

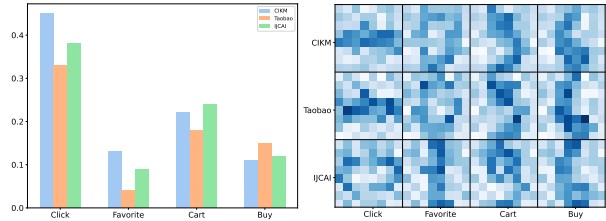

**Figure 4: Visualization of hard noise removal ratios (left) and soft denoising kernels (right) for different behaviors.**



**Figure 5: Comparison of Fourier spectral matrices with (left) and without (right) compactness regularization loss.**

we compute noise removal ratios for different behaviors identified by END4Rec and indicate that for behavior types with randomness, such as *clicking*, the corresponding sequence noise ratio is higher, while for behaviors with clear user preferences, such as *purchasing*, the noise ratio is relatively lower, which is consistent with practical expectations. (2). For the soft denoising process, we visualize the denoising kernels at the center of various behavioral channels to reveal their complex patterns. The comparison visualization reveals that the denoising kernels exhibit varying patterns across different datasets, while those with similar behaviors within the same dataset, such as *adding to cart* and *purchasing*, present similar denoising kernels. This finding indicates that END4Rec can effectively explore the correlations between different types of behaviors during the learning process. Besides, we also compare the Fourier spectrum matrices using and without compactness regularization in END4Rec. As demonstrated in Fig. 5, this figure shows the proposed compactness regularization function effectively sparse the representation space in the frequency domain, thus preventing overfitting to local information.

## 6 CONCLUSION

In this study, we aimed to address the challenges associated with mining user behavior sequences and reducing noise signals in multi-behavior sequential recommendation. To achieve this, we developed an Efficient Behavior Sequence Miner (EBM) that efficiently captures intricate patterns in user behavior while maintaining low time complexity and parameter count. Additionally, we introduced a Noise-Decoupling Contrastive Learning approach and a guided training strategy that combines the Hard Noise Eliminator and Soft Noise Filter techniques. These methods successfully eliminated noise from user behavior data. We conducted experiments on real-world datasets to evaluate the performance of our proposed method (END4Rec), which demonstrated its efficiency and effectiveness. For future work, we plan to further refine END4Rec by enhancing its capabilities for handling diverse types of noise signals.

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
