# OpenReview forum: "Efficient Noise-Decoupling for Multi-Behavior  Sequential Recommendation"
_ACM.org/TheWebConf/2024/Conference — TheWebConf24_

### Official Review · Reviewer_sXci · 2023-11-20

**Novelty:** 4
**Technical Quality:** 5

**Review:**

The paper addresses two challenges in multi-behavior sequential recommendation: (1) the   inefficiency of long sequences (2) noisy multi-behavior recommendation (e.g., click, add to bag, purchase). To address the first challenge, the paper proposes an efficient behavior sequence miner (EBM), a variation of the fast Fourier transform and an extension of the baseline FMLP. To address the second challenge, the paper proposes a hard noise eliminator, a soft noise filter and a noise-decoupling  contrastive learning framework. Experiments show that the proposed method outperform baselines. While the paper demonstrates the effectiveness of its approach through extensive experiments, there are notable concerns regarding the experimental setup's comparability with similar studies and the lack of detailed analysis on hyperparameter impacts.
Pros:
1.	The Behaviour-Aware Denoising module addresses both hard and soft noise and a contrastive loss for comparing denoised sequences and the original sequences.
2.	Comprehensive experiments show that the proposed method improvements prediction accuracy in multi-behaviour recommendation.

Cons:
1.	The paper integrates multiple components together to address two challenges. In the experiments, the overall performance is better than other baselines. However, it is not clear which components has the best contribution, the EBM or the denoising framework.  In the ablation study, a model with EMB only should be added.
2.	Experimental Result: The baselines includes DPT and ADT denoising models, which performs worse than the proposed model. In the DPT paper, ADT shows competitive performance, being only around 2% worse than DPT. However, in this paper, ADT performs significantly worse, showing more than a 30% decrease in performance compared to DPT. Any explanation for the implementation of those baselines?

**Questions:**

1.	In the DPT paper, ADT shows competitive performance, being only around 2% worse than DPT. However, in this paper, ADT performs significantly worse, showing more than a 30% decrease in performance compared to DPT. Any explanation for the implementation of those baselines?

**Reviewer Confidence:**

3: The reviewer is confident but not certain that the evaluation is correct

**Scope:**

4: The work is relevant to the Web and to the track, and is of broad interest to the community

---

### Official Review · Reviewer_4vPh · 2023-11-23

**Novelty:** 5
**Technical Quality:** 5

**Review:**

## summary:

This paper addresses challenges in multi-behavior sequential
recommendation systems, aiming to enhance performance by considering
various user behaviors and addressing the noise introduced by diverse
behavior data. The Efficient Behavior Sequence Miner (EBM) module is
introduced, employing fast Fourier transform to efficiently capture
complex user behavior patterns with reduced time complexity. Denoising
is approached through two modules: the Hard Noise Eliminator,
considering the relationship between noise and behaviors, and the Soft
Noise Filter, separating behaviors for effective noise removal. The
noise-decoupling contrastive learning technique is implemented to
preserve essential user interests while eliminating noise. A guided
training strategy, encompassing pretraining, hard noise comparison,
soft noise comparison, and final convergence, is proposed for
comprehensive model training.

## strengths:

- s1. The authors introduced a novel dual approach, implementing a
  two-level noise-decoupling strategy for sequential recommendation.

- s2. The proposed model was systematically compared against various sets
  of baselines, including:
  - General sequential recommendation models such as SASRec and BERT4Rec,
    Clrec and FEArec.
  - Multi-behavior recommendation methods.
  - Denoising methods specifically designed for recommendation tasks.

- s3. The evaluation demonstrated good consistency, with an average
  improvement ranging from 5% to 8% over all the baselines. These
  results were observed across three publicly available
  datasets. Additionally, the authors conducted thorough ablation
  studies to further analyze the effectiveness and robustness of their
  proposed approach.

## weaknesses:
- w1. Categorizing noise into discrete and continuous types is a
  simplification. Real-world noise in user behavior data can be much
  more nuanced, and these categories might not cover all the noise types
  effectively. This could lead to incomplete denoising in practical
  applications.

- w2. The section "5.1.4 Implementation Details" lacks important
  implementation details such as the optimizer used and the number of
  negative samples

**Questions:**

see weaknesses above

**Reviewer Confidence:**

3: The reviewer is confident but not certain that the evaluation is correct

**Scope:**

4: The work is relevant to the Web and to the track, and is of broad interest to the community

---

### Official Review · Reviewer_Ju3n · 2023-11-24

**Novelty:** 5
**Technical Quality:** 5

**Review:**

The paper studies noise-decoupling for multi-behavior sequential recommendation. It first identifies two challenges of the efficiency when user behavior sequences become very long in the short term and the noise of modeling user interest. To address the two challenges, it first develops the Efficient Behavior Sequence Miner (EBM) to efficiently capture intricate patterns in user behavior while maintaining low time complexity and parameter count. Then it designs hard and soft denoising modules for different noise types. Finally, it introduces a contrastive loss function along with a guided training strategy to contrast the valid information with the noisy signal in the data, and integrate the two denoising processes to achieve a high degree of decoupling of the noisy signal. It conducts extensive experiments on real-world datasets and demonstrates their effectiveness and efficiency.

Strengths:
1. The paper is generally well written and well organized.
2. The paper takes a holistic perspective on noise detection, considering the diversity of noise and providing new research directions from the angle of noise decoupling.
3. The authors propose an efficient behavior sequence miner (EBM) that effectively captures complex patterns in user behavior while maintaining low time complexity and parameter count.
4. The authors conduct extensive experiments on three real-world datasets and demonstrated the superior performance of the END4Rec method in multi-behavior recommendation. Furthermore, they validate the effectiveness of the module by intuitively demonstrating the complex relationship between noise and behavior through visual analysis.

Weaknesses:
1. Compared to end-to-end training, the training process of the model is a bit more complex which requires four stages.
Moreover, it is not clear if the performance comparison is fair since many other baselines only trains once. Combining Table 2 and Table 4, the proposed model using end-to-end training method (w/o opti in Table 4) shows insignificant advantages over DPT in Table 2. Therefore, the effectiveness of de-noising is suspicious to some degree.
2. Although the proposed EBM module reduces the parameter count through the Chunked Diagonal Mechanism, the model's overall complexity remains high due to multiple reuse of the module within a single model. As shown in Figure 2 and Algorithm 1, the END4Rec model may exploit EBM 6 times. Therefore, the total efficiency is also suspicious to some degree. Could you please check and display the real runtime in experiments?
3. Minor issues: there are some duplicate references (e.g., [2] and [3], [37] and [38], [44] and [45]).

**Questions:**

The visual analysis in Section 5.5 intuitively demonstrates the complex relationship between noise and behavior and yields some interesting findings. The paper mentions that "denoising kernels exhibit different patterns in different datasets, while denoising kernels with similar behaviors (e.g., adding to cart and purchasing) exhibit similar denoising kernels within the same dataset." However, from the presented Figure 4 (the right subfigure), some data points contradict this conclusion. For example, the denoising kernels for the "purchase behavior" across the three datasets appear similar to each other, and the denoising kernels for four behaviors in the CIKM dataset exhibit some similarity to that of the TaoBao dataset.. This may require further experimental analysis and clarification.

**Ethics Review Description:**

Non

**Reviewer Confidence:**

3: The reviewer is confident but not certain that the evaluation is correct

**Scope:**

4: The work is relevant to the Web and to the track, and is of broad interest to the community

---

### Official Review · Reviewer_WSoB · 2023-11-26

**Novelty:** 6
**Technical Quality:** 5

**Review:**

This paper addresses the challenges of efficiently utilizing user multi-behavior data and identifying noise data in sequential recommendation systems. To tackle these issues, the authors propose an Efficient Behavior Sequence Miner (EBM) module, which leverages frequency-domain operations for capturing complex user patterns while maintaining low computational cost. They also introduce a Behavior-Aware Denoising module, including components like the Hard Noise Eliminator and Soft Noise Filter, along with a guided training strategy to effectively remove noise from user behavior data. Strengths of the paper include:

1. The paper presents an innovative approach to noise-decoupling in sequential recommendation, emphasizing both hard and soft noise types. The proposed EBM module efficiently captures user behavioral patterns while minimizing model complexity, enhancing the robustness of denoising.

2. The paper conducts thorough experiments on real-world datasets, demonstrating the effectiveness and efficiency of the proposed END4Rec framework.

My primary concern regarding the paper is that the proposed model primarily addresses these issues from a mathematical perspective without providing corresponding explanations from the standpoint of the underlying principles of recommendation systems. It lacks insight into how different user behaviors reflect their preferences for items, making it challenging to grasp the model's underlying rationale. For instance, in the hard noise eliminator, the use of a Poisson distribution to "reflect the degree of user interest in various behaviors" seems less intuitive since different user behaviors are closely related to the target item, and using such a distribution to capture user interest in various behaviors may appear less justified.

Furthermore, it would be beneficial for the paper to specify the hyperparameter settings of the proposed model in the implementation details to enhance reproducibility. While hyperparameter studies are conducted, the manner in which these parameters are configured to achieve the reported results remains unclear. Additionally, the absence of code release makes it challenging for others to replicate the results.

Moreover, I noticed that the MB-CGCN model, published in WWW 2023, has shown significant improvements over other multi-behavior recommendation models. It would be valuable for the paper to acknowledge and compare its approach to MB-CGCN in the experiments section, providing a broader context for the evaluation.
Multi-Behavior Recommendation with Cascading Graph Convolution Networks. The Web Conference 2023

**Questions:**

Please explain how these components model user preferences based on their behaviors towards items, including the EBM, Hard Noise Eliminator, and Soft Noise Filter. This clarification would provide readers with a clearer perspective on how the model operates in capturing and denoising user behavior data for recommendation purposes.

**Ethics Review Description:**

N.A.

**Reviewer Confidence:**

4: The reviewer is certain that the evaluation is correct and very familiar with the relevant literature

**Scope:**

3: The work is somewhat relevant to the Web and to the track, and is of narrow interest to a sub-community

---

### Official Review · Reviewer_vQGM · 2023-11-26

**Novelty:** 5
**Technical Quality:** 5

**Review:**

This method utilizes a fast Fourier transform to achieve complex convolution operations with a time complexity lower than linear, effectively capturing complex patterns in user behavior sequences. A hard noise canceller and a soft noise filter were designed for different noise signals and user behavior denoised in discrete and continuous representation spaces. A contrastive loss function and a phased guidance training strategy were introduced to achieve sufficient decoupling of noise signals by comparing effective information and noise signals.

Strong points:
S1. This method was tested on three real-world datasets, and the results showed significant improvements in both indicators, demonstrating its effectiveness in the multi-behavior sequential recommendation tasks.
S2. This method reduces the model's time complexity and number of parameters by using multiplication operations in the frequency domain instead of convolution operations in the time domain, enabling it to process long multi-behavior sequences efficiently.
S3 This method considers the relationship between different types of noise signals and user behavior, designs an adaptive denoising module, and introduces strategies of contrastive learning and guided training to remove noise signals effectively.

Weak points:
W1. I suggest the authors elaborate on why multiplication operations in the frequency domain can effectively capture user behavior patterns.
W2. The information statistics in Table 1 only include users, items, and the number of interactions, with a relatively small amount of information. Suggest increasing details such as average sequence length and data partitioning.
W3. If integrating users' multi-behavior embedding information into sequence recommendation methods is helpful for recommendation effectiveness, it is recommended that the authors supplement relevant experiments.

**Questions:**

W1-W3.

**Reviewer Confidence:**

3: The reviewer is confident but not certain that the evaluation is correct

**Scope:**

4: The work is relevant to the Web and to the track, and is of broad interest to the community

---

### Decision · Program_Chairs · 2024-01-22

**Decision:**

Accept

**Comment:**

This paper introduces a method that employs fast Fourier transform for executing complex convolution operations more efficiently than linear time complexity, adeptly identifying intricate patterns in user behavior sequences. The reviewers broadly concur that the work is robust. The authors have successfully addressed the reviewers' concerns and inquiries. It is recommended that the authors incorporate this content into their discussion to enhance the robustness of the work further.